# Leveraging the Polymorphism of the Merozoite Surface Protein 2 (MSP2) to Engineer Molecular Tools for Predicting Malaria Episodes in a Community

**DOI:** 10.3390/ijms26115277

**Published:** 2025-05-30

**Authors:** Edgar Mutebwa Kalimba, Sandra Fankem Noukimi, Jean-Bosco Mbonimpa, Cabirou Mounchili Shintouo, Radouane Ouali, Mariama Telly Diallo, Antoine Vicario, Samuel Vandecasteele, Abenwie Suh Nchang, Lahngong Methodius Shinyuy, Mary Teke Efeti, Aimee Nadine Nsengiyumva Ishimwe, Aloysie Basoma Biryuwenze, Arsene Musana Habimana, Louis de Mont Fort Ntwali Mugisha, Sara Ayadi, Robert Adamu Shey, Rose Njemini, Stephen Mbigha Ghogomu, Jacob Souopgui

**Affiliations:** 1Laboratory of Embryology and Biotechnology, Department of Molecular Biology, Faculty of Science, Université Libre de Bruxelles, 6041 Gosselies, Belgium; edgar.kalimba@ulb.be (E.M.K.); sandra.fankem.noukimi@ulb.be (S.F.N.); mariama.telly.diallo@ulb.be (M.T.D.); antoine.vicario.108@hotmail.fr (A.V.); sara.ayadi@ulb.be (S.A.); 2Rwanda Malaria Research Lab, King Faisal Hospital Rwanda, Kigali P.O. Box 2534, Rwanda; furerebosco@gmail.com (J.-B.M.); naimeenadine2001@gmail.com (A.N.N.I.); aloysiebb1@gmail.com (A.B.B.); habimanaarsene79@gmail.com (A.M.H.); louisdemontfort30@gmail.com (L.d.M.F.N.M.); 3Department of Microbiology and Immunology, College of Medicine, Drexel University, Philadelphia, PA 19129, USA; cs3955@drexel.edu; 4Laboratory of Vector-Pathogen Biology, Proteomic Platform, Department of Molecular Biology, Université Libre de Bruxelles, 6041 Gosselies, Belgium; radouane.ouali@ulb.be; 5Department of Biomedical Sciences, Université de Mons, 7000 Mons, Belgium; samuel.vandecasteele@student.umons.ac.be; 6Department of Biochemistry and Molecular Biology, Faculty of Science, University of Buea, Buea P.O. Box 63, Cameroon; snabenwie@gmail.com (A.S.N.); lahngongmethodius@gmail.com (L.M.S.); sheynce@gmail.com (R.A.S.); stephen.ghogomu@ubuea.cm (S.M.G.); 7Department of Gerontology, Faculty of Medicine and Pharmacy, Vrije Universiteit Brussel, 1090 Brussels, Belgium; tekeefetimary@gmail.com (M.T.E.); rose.njemini@vub.be (R.N.)

**Keywords:** malaria surveillance, MSP2 polymorphism, IgG3 response, outbreak prediction

## Abstract

Malaria remains a significant public health challenge, particularly in endemic regions. The extensive genetic diversity of *Plasmodium falciparum* (*Pf*) complicates outbreak prediction and transmission control. One of its most polymorphic markers, merozoite surface protein 2 (MSP2), presents a potential target for molecular surveillance. This cross-sectional study, conducted at King Faisal Hospital Rwanda (KFHR) from October 2021 to June 2023, assessed MSP2’s utility in malaria prediction. *Pf*MSP2 was sequenced, and selected amplicons were cloned, expressed in bacteria, and purified. These antigens were tested against sera from malaria patients and geographically diverse healthy individuals, with complementary surveys contextualizing serological findings. Of the 75 processed monoallelic clinical isolates, 3D7 strains predominated over FC27. Three MSP2-derived biomarkers were produced, eliciting significantly low IgG responses in malaria patients and Belgian controls, but a complex pattern emerged in healthy individuals, with significant differences between Rwandan and Cameroonian samples. IgG3 was the predominant subclass in individuals with high IgG responses. Notably, Rwandan individuals with weak humoral responses to the tested antigens but also other with high responses experienced malaria episodes in the subsequent year. These findings highlight MSP2 polymorphism as a valuable tool for malaria surveillance and outbreak prediction. Integrating genotyping and serology could enable precise, community-specific malaria risk assessments, strengthening control strategies.

## 1. Introduction

Malaria is a life-threatening disease caused by *Plasmodium* parasites, transmitted to humans by infected female *Anopheles* mosquitoes. Globally, over 263 million people in 85 endemic countries were infected in 2023, with nearly half of the world’s population at risk. The most vulnerable groups include children under five, pregnant women, travelers, and immunocompromised individuals. Malaria deaths in 2023 were estimated at 597,000, with over 94% of cases and 95% of deaths occurring in the WHO African Region [1]. Among the five *Plasmodium* species that cause malaria, *Plasmodium falciparum* is the most virulent, responsible for over 90% of cases and malaria-related deaths [2].

To evade the human immune system, *P. falciparum* undergoes antigenic variation, producing multiple clones that recombine during sexual reproduction in the mosquito gut [3,4]. This genetic recombination generates diverse parasite strains with varying levels of drug resistance, transmission efficiency, and virulence [5]. The infection outcomes range from asymptomatic carriage to severe disease, influenced by host immunity, parasite genetics, and environmental factors [6]. Asymptomatic individuals in endemic areas exhibit partial immunity, often carrying the parasite without symptoms but contributing to transmission [7,8,9,10]. These individuals typically harbor higher gametocyte densities, enhancing parasite spread [11]. Understanding their potential to develop symptomatic malaria is crucial for improving community-specific risk assessments and control strategies.

The polymorphic genes encoding the *P. falciparum* merozoite surface proteins 1 and 2 (MSP1 and MSP2), as well as the apical merozoite antigen 1 (AMA-1), have been used to study allelic diversity [12,13], and both gene families are dominant targets of immune response [14,15]. The extensive genetic diversity of *P. falciparum* limits the effectiveness of acquiring protective immunity to malaria, and this has been a major obstacle in the development of a vaccine for malaria [16]. The reports indicate that the locus of the *msp2* gene is extremely polymorphic [17] and has two main allelic families, 3D7 (D)-type and FC27 (F)-type [18]. Serological characterization of immune responses to antigens derived from *msp2* gene polymorphic regions can determine immune tolerance against *P. falciparum* strains in an endemic community. An individual may exhibit partial immunity to *P. falciparum,* depending on the strain that the individual is exposed to. Establishing a biomarker for the polymorphic genes of *P. falciparum* that give rise to allelic diversity may be used to determine the immune responses of human hosts against *P. falciparum* strains.

Hence, in this study, we investigated the allelic diversity of *P. falciparum* in Kigali, Rwanda, and assessed the immune responses of symptomatic, asymptomatic, and European control individuals against antigens of the FC27 and 3D7 polymorphic regions, followed by surveys to contextualize the serological findings. Our findings underscore the MSP2 polymorphism as a valuable biomarker for malaria surveillance and outbreak prediction, offering critical insights for improved control strategies.

## 2. Results

### 2.1. Study Characteristics

Between October 2021 and June 2023, genomic DNA was extracted from 240 blood samples collected from individuals presenting with malaria symptoms at King Faisal Hospital Rwanda, Kigali, and associated centers. *P. falciparum* infection was confirmed microscopically and further validated via PCR targeting the *Pfmsp2* gene. Samples were then sequenced using the Sanger method.

Among these, 75 samples were monoallelic and had corresponding serum samples from the same patients. This subset of 75 clinical isolates, consisting of 35 females and 40 males aged between 1 and 60 years (Table 1), was used for further investigation.

Additionally, between July 2022 and March 2023, serum samples were collected from healthy individuals in Bugesera, Rwanda, and Bafoussam, Cameroon. As controls, 26 serum samples from European individuals, previously collected in a separate study [19], were also included in the analysis.

### 2.2. The Polymorphism of the Pfmsp2 Gene from KFHR Clinical Isolates Revealed 3D7 Strains Predominant over FC27

As illustrated in Figure 1A, the alignment of the FASTA sequences from clinical isolates with mono-infections revealed gaps and sequence heterogeneity, reflecting the polymorphic nature of *Pfmsp2*. To mitigate the effects of potential sequencing artifacts, the first 50 base pairs (bp) and the last 20 bp of each sequence, which may have contained non-specific mutations due to technical limitations, were systematically removed.

The genetic tree representing allelic distances, shown in Figure 1B, suggests a predominance of *Pf*3D7 strains over FC27. Following the previously reported structural annotation of *Pf* MSP2 proteins [20], four main clusters were identified within the FC27 family and two within the 3D7 family (Figure 1B).

### 2.3. Cloning, Expression, and Characterization of MSP2 Selected Antigens: SM3, WB7, and SM68

The sequence alignment analysis enabled the selection of three specific polypeptides based on their sequence divergence and antigenicity within the polymorphic region of *Pf*MSP2. This selection included two polypeptides from the FC27 family, designated SM3 and WB7, and one from the 3D7 family, referenced as SM68. The corresponding translated DNA sequences are presented in Figure 2A. Amino acids in the N-terminal-conserved region are highlighted in blue, while the remaining protein sequences in black indicate amino acid variations that characterize the polymorphism and distinguish the parasite alleles.

Subsequently, the PCR amplicons of interest were cloned into the pET-30a (+) vector using the *Nde*I and *Xho*I restriction enzymes, as illustrated in Figure 2B. The resulting recombinant plasmids were used to transform *Escherichia coli* BL21 bacterial competent cells. As shown in Figure 2C, the pattern of IPTG-induced and non-induced transformants did not display any additional bands indicative of the expected recombinant polypeptides. However, despite the absence of visible bands in Coomassie-stained gels for all three target proteins, His-tagged purification confirmed the successful expression of the antigens by the IPTG-induced bacteria. Western blotting using anti-His-tag antibodies further validated this expression, revealing bands corresponding to the expected sizes of the MSP2 antigens (Figure 2D). Regarding WB7, the band is not visible on the SDS-PAGE due to low expression levels and the limited sensitivity of Coomassie staining. However, successful expression was confirmed by Western blot detection of the protein using an anti-His antibody.

### 2.4. Serum Samples from Malaria Patients Respond Differentially to the Candidate Biomarkers SM3, WB7, and SM68

A cohort of serum samples from KFH-malaria patients was analyzed to assess their reactivity to bacterially produced MSP2-derived antigens. Each candidate biomarker is represented on the allelic tree (Figure 3A), with a color-coded system used to differentiate the various sera (Figure 3B). ELISA analysis demonstrated significantly lower overall humoral responses to the antigens.

Notably, SM3 and WB7 consistently exhibited similar response patterns, reflecting their shared allelic cluster. As indicated by the discontinuous lines, some serum samples with low to very low responses to SM3 and WB7 displayed high reactivity to SM68, suggesting the prior exposure of the corresponding patients to a distinct *P. falciparum* strain (blue line). Conversely, sera with high levels of anti-SM3/WB7 antibodies showed low to very low responses to SM68 (brown line), supporting the interpretation of differential parasite strain exposures within the population.

### 2.5. The Candidate Biomarkers SM3, WB7, and SM68 Effectively Discriminate Sera from Inhabitants of Geographically Distinct Regions

The ability of the candidate biomarkers SM3, WB7, and SM68 to predict new malaria episodes was investigated by examining the humoral response of healthy individuals living in geographically distinct regions. First, we provided evidence that these antigens could significantly differentiate serum samples from malaria patients living in Kigali and those from healthy Rwandans in Bugesera, located in the Eastern Province (Figure 4A).

Next, Western blot analyses using pooled serum samples from healthy individuals in Rwanda, Cameroon, and Belgium revealed strong differential responses to the candidate biomarkers, as indicated by the presence or absence of bands with varying signal intensities. To corroborate these findings, individual serum samples were subsequently tested. Indeed, the sera from Rwanda, where the different *msp2* alleles used for cloning originated, exhibited the strongest response, followed by samples from Cameroon. In contrast, all samples from Belgium tested negative for the three antigens (Figure 4C).

### 2.6. Sera from Healthy Rwandans Living in Bugesera Exhibited Similar Response Patterns to the SM3, WB7, and SM68 Candidate Biomarkers

To investigate the possibility that one of the engineered antigens could be best recognized by serum samples from inhabitants living in a malaria hotspot in Rwanda, the produced recombinant proteins were analyzed by ELISA using the same cohort of sera. In contrast to the humoral response of malaria patients shown above (Figure 3B), no significant difference was detected in the response patterns (Figure 5A). Comparison between sera from healthy and malaria patients revealed a systematic poor response of patient serum samples to all the antigens (Figure 4A). Using the color code, the high or low response patterns were conserved for all three candidate biomarkers (Figure 5B). This conserved reactivity supports the reproducibility of the assay and the consistent recognition of the antigens across different exposure levels.

Next, follow-up surveys were conducted a year later to investigate the impact of immunity levels on the prevalence of malaria episodes in the Bugesera population. Despite the varying response levels to the three candidate biomarkers, no correlation between immunity levels and the prevalence of malaria episodes was observed. This lack of correlation could not be entirely explained by the movement of participants to other regions of Rwanda or by their hosting of relatives from other cities within the country (Figure 5C). These observations highlight the complexity of malaria transmission dynamics and suggest that additional environmental or behavioral factors may modulate individual susceptibility beyond measurable humoral responses.

### 2.7. The High Humoral Responses to the Three Candidate Biomarkers Are Predominantly Enriched in the IgG3 Antibody Subclass

To determine which of the four IgG antibody subclasses predominated in the sera that reacted strongly with the three candidate biomarkers, a pooled serum sample was first prepared and subjected to serial dilutions. The response of each antigen to the dilution series was assessed, with the 1:500 dilution providing the best discrimination among the four IgG subclasses (Figure 6A). This dilution was therefore used in subsequent assays to evaluate subclass-specific reactivity.

Next, individual serum samples exhibiting high responses to the three candidate biomarkers were used to probe each antigen using ELISA assay. A color-coding system was applied to differentiate the sera, and a consistent pattern was observed for all the tested MSP2-derived recombinant proteins. As shown in Figure 6B, the IgG3 antibody subclass was predominantly enriched in the reactive sera, suggesting that this subclass plays a central role in mediating humoral immunity in the context of *P. falciparum* and human–host interactions.

## 3. Discussion

### 3.1. Allelic Variation and Malaria Episode Prediction

The allelic variation of *P. falciparum* parasites, particularly in genes encoding surface antigens, plays a significant role in predicting malaria episodes. Allelic diversity allows the parasite to evade host immune responses [20], contributing to the complexity of malaria pathogenesis and complicating the prediction of future episodes. Continuous monitoring of parasite genetic diversity, coupled with serological data, can strengthen malaria surveillance systems and enable timely intervention strategies. Integrating molecular and immunological markers may thus enhance early warning capabilities and inform targeted control measures.

In this study, we identified, cloned, expressed, purified, and characterized three MSP2-derived candidate biomarkers from *P. falciparum* strains isolated from malaria patients at King Faisal Hospital in Kigali. These antigens were selected for their representativeness, as they correspond to some of the most prevalent MSP2 alleles detected in our cohort (Figure 1A), covering both major allelic families, 3D7 and FC27. In addition to their frequency, their selection was supported by high predicted antigenicity and confirmed immunoreactivity in ELISA assays. Indeed, they were tested against sera from malaria patients and geographically diverse healthy individuals, with complementary surveys contextualizing the serological findings. Among 240 processed clinical isolates, 3D7 strains predominated over FC27. Three MSP2-derived biomarkers were produced, eliciting significantly low IgG responses in malaria patients and Belgian controls. However, a complex response pattern was observed in healthy individuals, with significant differences between Rwandan and Cameroonian samples. IgG3 emerged as the predominant subclass in individuals with high IgG responses. Interestingly, some Rwandan individuals with weak humoral responses to the antigens, as well as others with strong responses, experienced malaria episodes in the subsequent year.

### 3.2. Distribution of Plasmodium falciparum MSP2 Allelic Families and Biomarker Engineering

PCR amplification of the *Pfmsp2* gene followed by sequencing was used to determine the relative abundance of 3D7 and FC27 allelic variants in clinical isolates from patients at King Faisal Hospital, Kigali, between October 2021 and June 2023. Since this cohort included referred patients with potential treatment failures or inadequate management from other health centers, the observed 3D7 to FC27 ratio of 2:1 may not accurately reflect the overall genetic diversity in Kigali or Rwanda. High transmission settings are typically associated with greater genetic diversity and a more balanced ratio between 3D7 and FC27 clusters [13,21,22]. However, in a low transmission context like Kigali during the study period, reduced parasite diversity may explain the dominance of the 3D7 family (Figure 1A and Figure 3A).

To investigate MSP2 antigenic diversity for candidate biomarker engineering, three allelic sequences of interest from both the 3D7 and FC27 families were selected based on MSP2 sequence alignment and in silico epitope prediction. Surprisingly, only three recombinant proteins (SM3, WB7, and SM68) out of the seven cloned variants were successfully expressed and purified using *E. coli* BL21 and Ni-NTA affinity chromatography columns (Figure 2). The failure to express certain MSP2 constructs can likely be attributed to the intrinsic biochemical properties of MSP2 proteins, which are known to be highly polymorphic and intrinsically disordered. These features often render the proteins unstable and prone to aggregation, particularly when heterologously expressed in *E. coli*. Such technical limitations are well documented for disordered antigens and may lead to poor solubility or degradation during expression and purification. Hence, *E. coli* is suitable for expressing only a small proportion of MSP2 proteins. These challenges restricted the number of variants that could be produced. Our findings are consistent with previous reports from a large-scale study involving 1000 *Plasmodium falciparum* open reading frames (ORFs), in which approximately 33.7% (337 genes) were successfully expressed in *E. coli*, but only 6.3% yielded soluble proteins [23]. However, the successfully expressed antigens are representative of the two major allelic families, as shown in Figure 1 and Figure 3A.

### 3.3. Differential Serum Responses to Candidate Biomarkers

Analysis of host antibody responses to different MSP2-specific alleles provides valuable insights into predicting susceptibility to malaria episodes. Variants of MSP2 antigens can evade recognition by pre-existing antibodies, thereby compromising the effectiveness of the host immune response and increasing the likelihood of symptomatic infection. Consistent with this, our findings revealed markedly low humoral responses to all three candidate biomarkers in malaria patients (Figure 3B). This suggests limited recent exposure to the specific *P. falciparum* strains from which the biomarkers were derived. Notably, a few serum samples with high responses to SM3 and WB7 (FC27 family) showed low responses to SM68 (3D7 family), and vice versa, supporting the existence of allele-specific humoral responses.

Comparing the malaria-specific humoral response of symptomatic patients to that of healthy individuals in Bugesera revealed a significant difference. The sera from healthy individuals displayed heterogeneous responses to the candidate biomarkers, reflecting differential past exposure to distinct *P. falciparum* strains (Figure 4A). The sera from Cameroonian individuals showed even lower responses to the biomarkers, underscoring the variation in parasite genetic architecture across regions. In contrast, Belgian sera exhibited no detectable response, reaffirming the lack of malaria exposure.

### 3.4. MSP2-Mediated Genetic Diversity and Infection Risk

The genetic diversity of *P. falciparum* significantly influences infection risk in immune populations. In regions with high malaria transmission, repeated exposure induces partial immunity. However, the parasite’s genetic variability [24] can undermine this immunity, maintaining a persistent infection risk. Gene conversion events contribute to the diversity of surface protein genes, facilitating immune evasion. As a result, individuals with acquired immunity may still be vulnerable to infection by genetically distinct parasite strains.

Populations with prolonged exposure may develop broader immune responses capable of recognizing a wider range of parasite antigens. This is exemplified by a subset of healthy Rwandan individuals exhibiting high IgG responses (Figure 5A, red rectangle box). Despite their strong humoral responses, some participants experienced malaria episodes in the subsequent year. Survey data revealed that these individuals had hosted visitors from malaria-endemic regions or traveled to other cities within Rwanda, suggesting exposure to diverse parasite strains and the risk of reinfection.

However, the cross-sectional design of our study limits causal interpretations of the association between MSP2 polymorphism and future malaria episodes. Although follow-up surveys were conducted to document subsequent cases, the absence of systematic data on potential confounding factors, such as individual mobility or local vector control interventions, represents a limitation. Future longitudinal studies integrating and applying these variables will be essential to confirm the predictive value of MSP2 diversity in malaria risk assessment.

While our results indicate that MSP2 polymorphism may be associated with malaria susceptibility, we did not observe a statistically significant correlation between the levels of antibodies directed against the selected MSP2 variants and the incidence of subsequent malaria episodes. This limitation suggests that antibody responses alone may not fully reflect protective immunity or predict clinical outcomes, particularly in settings with high antigenic diversity and fluctuating transmission. It also underscores the need to consider other immune effectors and host-related factors that may modulate susceptibility.

The multiplicity of infection (MOI) provides valuable insight into host exposure to diverse *P. falciparum* strains. A higher MOI reflects repeated encounters with antigenically distinct parasites, which may promote broader immune responses and partial protection. In contrast, individuals, especially children with limited exposure (low MOI) remain vulnerable to symptomatic infections. In our study, low IgG responses in malaria patients likely indicate limited prior exposure to the selected MSP2 variants.

### 3.5. IgG Subclass Responses and Malaria Immunity

Naturally acquired immunity to *P. falciparum* malaria is predominantly mediated by the IgG1 and IgG3 subclasses, which possess cytophilic properties and facilitate parasite clearance [25]. Studies analyzing IgG responses in African children vaccinated with RTS,S/AS01E also emphasized the importance of IgG1 and IgG3 [19]. Consistently, our candidate biomarkers revealed IgG3 as the most enriched subclass (Figure 6B), underscoring its potential protective role. Similar findings have reported associations between cytophilic IgG1 and IgG3 responses and a reduced risk of malaria [26].

## 4. Materials and Methods

### 4.1. Ethical Consideration

The project was approved by the University of Buea Faculty of Health Science Institutional Review Bord in Cameroon (2022/1591-01/UB/SG/IRB/FHS), the Rwandan National Ethics Committee (120/RNEC/2022). All individuals who voluntarily decided to participate in this study were given informed consent forms, which they signed after being explicitly informed about the project. The privacy of participants was safeguarded throughout data collection, processing, and reporting.

### 4.2. Study Site and Population

To investigate *P. falciparum* allelic frequency in a cosmopolitan town in Rwanda, outpatients with fever (body temperatures greater than 37.5 °C) and/or malaria-related symptoms within three days before seeking medical care, and the presence of a *P. falciparum* positive blood smear, were recruited in selected health facilities in Kigali (Figure 7). To determine immune responses to antigens from *msp2* gene polymorphic regions, healthy individuals were recruited from Bugesera in Rwanda and Bafoussam in Cameroon. European individuals who have never visited malaria-endemic regions were recruited as controls.

### 4.3. Parasite DNA Extraction

First, 2 mL of blood samples was collected from each participant and stored in a DNA/RNA shield transport medium (Zymo Research, Tustin, CA, USA). Parasite genomic DNA was extracted from whole blood using a Maxwell 16 RSC robot (Promega, Madison, WI, USA) and Maxwell RSC Blood Kit (Promega, Madison, WI, USA).

### 4.4. Plasmodium falciparum Genotyping

To genotype different forms of *P. falciparum* in the blood samples from symptomatic malaria patients, the polymorphic repetitive regions of the *msp2* gene were amplified by PCR. The MSP2 primers present in Table 2 were designed to anneal to the conserved region of the gene, permitting the amplification of both the 3D7 and FC27 strains.

PCR amplification was performed using a total of 25 µL of reaction volume. The mix contained 2× of Go-Tag Green Master Mix (Promega), 2 µL of genomic DNA, 1.5 µL of 10 µM each of forward and reverse primer (Integrated DNA Technologies, Leuven, Belgium), and 7.5 µL of nuclease-free water. The DNA amplification was carried out under the following standard conditions: initial denaturation: 95 °C for 3 min, denaturation: 94 °C for 30 s, annealing: 55 °C for 60 s, polymerization: 72 °C for 60 s, and final extension: 72 °C for 5 min. A 35 cycle PCR reaction was performed.

The PCR products were separated on a 2% agarose gel and viewed under ultraviolet (UV) light using the molecular imager GelDocTM XR+ (BIORAD, Hercules, CA, USA).

### 4.5. DNA Sequencing

The nested PCR product was purified using the Wizard SV Gel/PCR Cleanup System kit (Promega) according to the manufacturer’s protocol. Purified DNA products were quantitated by Nanodrop, and their concentrations were normalized to 20 ng/µL. Sequencing tubes containing equal volumes of normalized DNA product and 5 µM of corresponding forward or reverse nested primer (Table 2) were prepared for Sanger sequencing (Eurofins Genomics, Ebersberg, Germany).

### 4.6. Cloning of msp2 Polymorphic Gene Regions, Expression, and Purification of Antigens

The nested PCR products from patients SM3, SM68, and WB7, which were predicted by Vaxijen v2.0 (http://www.ddg-pharmfac.net/vaxijen/VaxiJen/VaxiJen.html, accessed on 15 March 2023) to be most antigenic, were double digested with *Nde*I and *Xho*I, and a pET-30a(+) vector was also treated with the same restriction enzymes to linearize the supercoiled plasmid. The digests (PCR product and linear pET-30a(+) vector) were mixed and treated with T4 ligase for the ligation of the construct into the bacterial expression pET-30a(+) vector. Ligation was confirmed by sequencing the recombinant vector that was later used to transform *E. coli* BL21(DE3) chemically competent cells (Sigma, St. Louis, MO, USA) for protein expression [27]. The transformed cells were cultured in Terrific Broth medium (Sigma, St. Louis, MO, USA), supplemented with 50 µg/mL kanamycin, at 37 °C with a shaking speed of 180× *g*. When the OD600 of the cells reached 0.7, 0.5 mM isopropyl-β-D-1-thiogalactopyranoside (IPTG) was added for induction.

The cells were harvested after 3 h by centrifugation at 3578× *g* for 10 min. The harvested cells were resuspended in lysis buffer (50 mM NaH_2_PO_4_, 300 mM NaCl, 10 mM imidazole, pH 8.0) and frozen overnight at −20 °C. Thereafter, the sample was placed in an ice-water bath and lysed by sonication in short pulses of 30 s at 60% amplitude for 30 min. Centrifugation at 20,000× *g* for 20 min was then performed, and the supernatant was loaded on Ni(II)-immobilized metal-affinity chromatography columns (GE Healthcare, Diegem, Belgium) and eluted using 250 mM imidazole following standard protocol [27]. The purified proteins were resolved on SDS-PAGE, and its identity confirmed by Western blot using anti-His monoclonal antibodies (Thermo Fisher Scientific, Merelbeke, Belgium), as previously described [28]. The concentrations of purified proteins were determined using Pierce™ 660 nm Protein Assay (Thermo Fisher Scientific, Merelbeke, Belgium), using a series of bovine serum albumin (BSA) solutions as protein concentration standards.

### 4.7. Western Blotting

Initially, 10 µg of proteins was denatured in Laemmli buffer (0.125 M Tris-HCl, 4% SDS, 20% glycerol, 10% 2-mercaptoethanol, 0.004% bromophenol blue) and subsequently loaded onto a 15% SDS-PAGE gel. After electrophoretic separation, the proteins were transferred onto a nitrocellulose membrane (Thermo Fisher, Merelbeke, Belgium) by electroblotting for 75 min at 150 V at 4 °C, using a transfer buffer containing 0.25 M Tris, 200 mM glycine, and 20% methanol. The membrane was then blocked overnight in TBS (20 mM Tris, 150 mM NaCl, pH 7.6) containing 5% milk and 0.1% Tween 20. It was subsequently incubated for 1 h with an HRP-conjugated anti-His antibody (Proteintech, Manchester, UK) in TBS supplemented with 2.5% milk and 0.05% Tween 20. The membrane was washed five times with TBS containing 2.5% milk and 0.05% Tween 20 to remove unbound antibodies, followed by three additional washes with the same solution and two final washes with TBS alone. Bound antibodies were detected by chemiluminescence, and the emitted light was captured using a CCD camera (Odyssey^®^ Fc, Bad Homburg, Germany).

### 4.8. Serological Assessment of the Expressed Antigens

The total IgG responses to antigens from different *P. falciparum* alleles were investigated by indirect ELISA using healthy and non-infected *P. falciparum* sera. Optimal antigen/antibody concentrations were determined by the checkerboard titration method. Maxisorp 96-well microtiter plates (Nunc, Roskilde, Denmark) were coated with 50 µL of 1 μg/mL of the purified antigens at 37 °C for 2 h. Plates were washed three times with wash buffer (PBS + 0.05% Tween 20) and blocked overnight with PBS supplemented with 150 µL of 3% Bovine Serum Albumin (BSA) (Sigma, St. Louis, MO, USA). The plates were then washed and incubated with the different serum samples at a dilution of 1:8000 for 2 h at room temperature. After washing the plates, they were incubated with goat anti-human IgG (Fc Specific) peroxidase conjugate (Sigma, St. Louis, MO, USA) at a dilution of 1:5000 for 1 h 30 min at room temperature. After a final wash, 3,3′,5,5′-tetramethylbenzidine (TMB, Sigma, St. Louis, MO, USA) was added for 5 min at room temperature. The reactions were stopped with 1 M hydrochloric acid after which the absorbance was read at 450 nm using the Berthold microplate reader (Berthold, Bad Wildbad, Germany). All antibody dilutions were conducted in a blocking buffer (PBS supplemented with 3% BSA).

For the IgG subclasses’ titration, a serial dilution of a pool of 10 serum samples from high response to the candidate biomarkers was used, followed by incubation with IgG1 to IgG4 subclasses, and revealed as described above. For the individual serum analysis, a dilution of 1:2000 was used.

### 4.9. Qualitative Analysis of P. falciparum Infection

Healthy individuals from Bugesera, Rwanda, who reacted strongly (OD ≥ 2.5) or weakly (≤0.1) with the antigens were invited for a follow-up study in May 2024 to investigate their previous frequency of malaria episodes using a questionnaire which was translated to Kinyarwanda by a clinical researcher fluent in English at King Faisal Hospital Rwanda in Kigali.

### 4.10. Data Analyses

Categorical variables were presented as proportions, and numerical variables as medians with interquartile ranges (IQRs) or means with standard deviations (SDs), as appropriate. FC27 and 3D7 allelic frequencies were calculated as the proportion of each msp2 allelic family relative to the total number of alleles detected. The number and size of bands per sample indicated the minimum number of clones per positive sample, with single infections defined as samples with one band and multiple infections as those with two or more bands. The multiplicity of infection (MOI) was estimated by counting the number of different alleles detected per sample; the mean MOI was calculated by dividing the total number of MSP2 genotypes by the number of positive samples for each marker.

For group comparisons, chi-square tests were used for categorical variables, while Student’s *t*-test or one-way ANOVA were applied to compare means across groups, depending on the number of categories. For ELISA data, differences in antibody levels among groups were assessed using non-parametric tests (Mann–Whitney *U* test or Kruskal–Wallis test, as appropriate). Logistic regression (univariate and multivariate) was used to evaluate associations between independent variables and polyclonal infections, with odds ratios (ORs) and 95% confidence intervals (CIs) reported.

Receiver operating characteristic (ROC) curve analysis was used to evaluate the discriminative ability of total IgG responses, and the area under the curve (AUC) was computed using the trapezoidal method. All analyses were performed using GraphPad Prism 5, and adjusted *p*-values < 0.05 were considered statistically significant to control for the type I error in multiple testing contexts.

## 5. Conclusions

Our findings underscore the intricate relationship between *P. falciparum* genetic diversity and the ongoing challenges of malaria control in populations with varying immunity levels. Developing biomarkers based on MSP2 genetic diversity could enhance malaria episode prediction and serve as a valuable tool for global malaria control strategies. Future longitudinal studies that incorporate variables such as individual mobility and local vector control interventions will be essential to validate the predictive value of MSP2 diversity in malaria risk assessment.

## Figures and Tables

**Figure 1 ijms-26-05277-f001:**
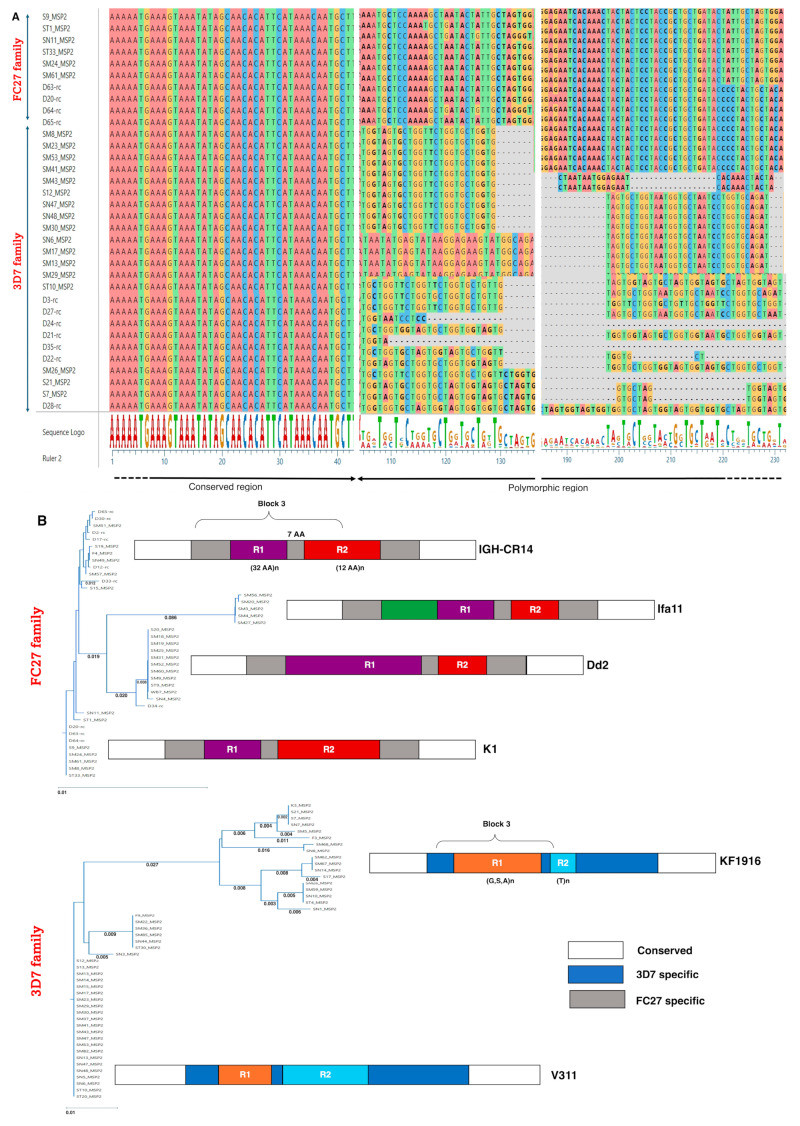
Comparative sequence analysis of the *P. falciparum* msp2 gene from patient isolates. (**A**) Illustration of a part of alignment from 75 msp2 gene sequences showing polymorphic features of interest. *P. falciparum* patient isolates were amplified by PCR and sequenced using the Sanger method. The alignment highlights conserved and polymorphic regions relative to the msp2 reference sequence. Below, a sequence logo represents the nucleotide distribution at each position, illustrating the probability of nucleotide occurrence across the analyzed sequences. (**B**) Cladogram illustrating the classification of *P. falciparum* isolates into FC27- and 3D7-like families based on msp2 gene sequences. The structural organization of the variable region is shown for representative sequences. Block 3 consists of two repeats: units (R1 and R2), separated by a non-repeat region. Based on the repetition patterns of the variable motifs, the FC27 family was further subdivided into four groups (IGH-CR14, Ifa11, Dd2, and K1), while the 3D7 family was divided into two groups (KF1916 and V311) [17].

**Figure 2 ijms-26-05277-f002:**
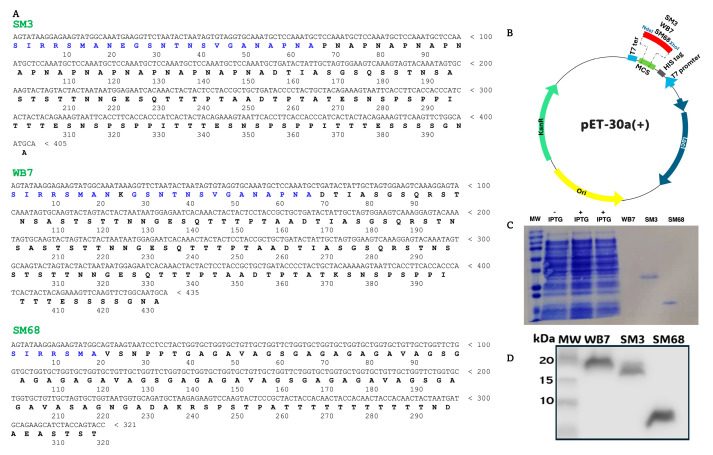
Expression and characterization of MSP2 selected antigens: SM3, WB7, and SM68. (**A**) Translated sequences originating from the different MSP2 variants of distinct isolates. (**B**) Schematic representation of the cloning strategy for the selected antigen sequences. Inserts were introduced into the pET-30a (+) expression vector between the *Nde*I and *Xho*I sites, downstream of the His-tag sequence. The resulting constructs were used to transform *E. coli*. (**C**) SDS-PAGE analysis of the expression of the three antigens following IPTG induction. (**D**) Western blotting analysis using an anti-His antibody to detect the three antigens expressed in *E. coli* following Ni-NTA column purification.

**Figure 3 ijms-26-05277-f003:**
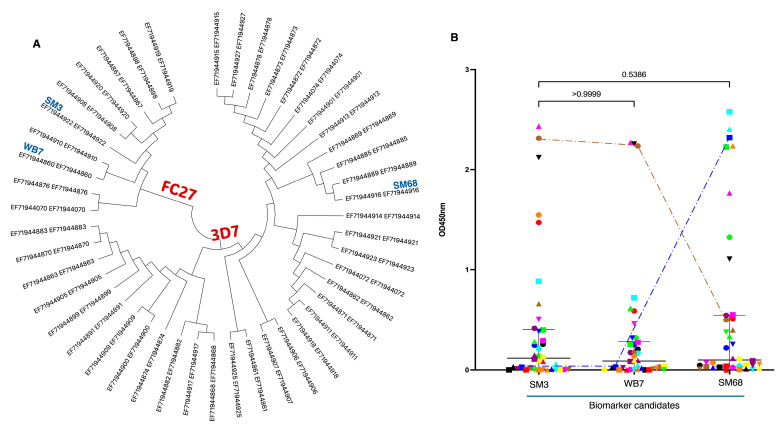
Differential response of malaria patient serum samples to the candidate biomarker candidates. (**A**) Cladogram analysis of isolates from patients based on their MSP2 sequence, showing that some patients are infected with strains from the FC27 family, while others are infected with strains from the 3D7 family. The genetic tree was established using the Sanger barcode references allocated to the different *Plasmodium* parasite strains. The samples from which our biomarker candidates derived are indicated. (**B**) Analysis of the reactivity of patient serum samples to the three selected antigens (SM3, WB7, and SM68) as biomarkers, assessed by ELISA. Everyone is presented by specific color code. The data demonstrate a differential, family-specific reactivity. Serum samples that strongly react to FC27 family antigens (SM3 and WB7) show little or no reactivity to the SM68 antigen from the 3D7 family, and vice versa. This is illustrated by the differential reactivity observed in the two selected patients, for whom reactivity to the three antigens is indicated by dashed lines. The results are expressed as mean ± SD. The results were compared by *t*-test, and the *p*-value for each pairwise comparison is provided.

**Figure 4 ijms-26-05277-f004:**
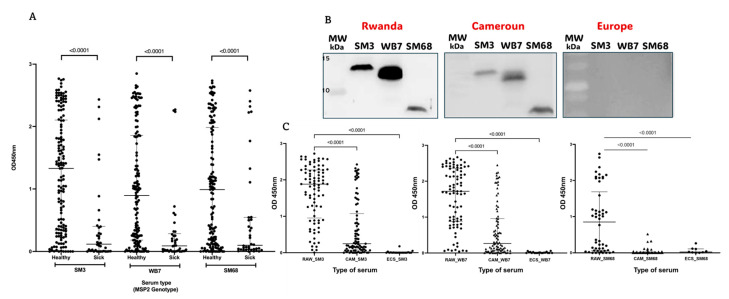
Differential response of serum from healthy individuals and malaria patients to selected biomarker candidates. (**A**) Serum samples from both malaria patients and healthy individuals were tested for their reactivity to the three biomarker candidates using ELISA. (**B**) Western blot analysis of the three recombinant antigens using serum from individuals residing in endemic regions of Rwanda, Cameroon, and European controls who have never traveled to endemic areas. (**C**) Reactivity of serum samples from individuals residing in three geographically distinct regions to the SM3 and WB7 biomarkers was evaluated by ELISA. The Elisa results were expressed as mean values ± SD, with statistical comparisons performed using one-way Anova test. The *p*-values for the comparison of reactivity between compared groups are indicated.

**Figure 5 ijms-26-05277-f005:**
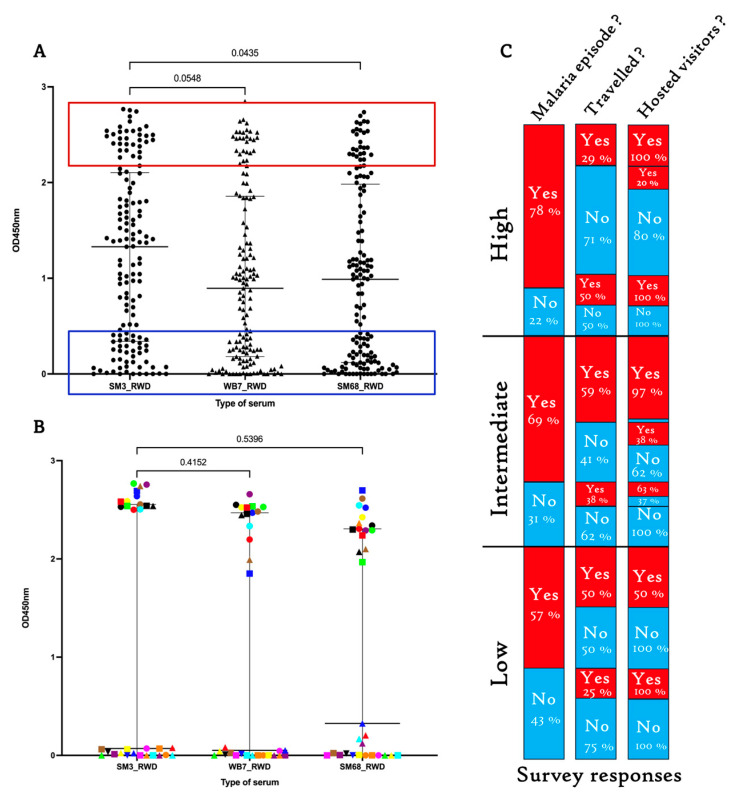
Humoral response of *P. falciparum* asymptomatic individuals to selected biomarker candidates, and associated survey (**A**) serum samples from healthy individuals were tested for their reactivity to the three candidate biomarkers using ELISA. (**B**) A set of sera marked by a specific color code, with high (OD > 2.5) and low (OD < 0.5) humoral responses, were retrieved, and their response pattern to the three antigens compared, highlighting a conserved response independent of the tested marker. The Elisa results were expressed as mean values ± SD. The *p*-values for the comparison of reactivity between compared groups are indicated. (**C**) Survey response targeting individuals with different humoral response to the candidate biomarkers. The main survey questions are indicated.

**Figure 6 ijms-26-05277-f006:**
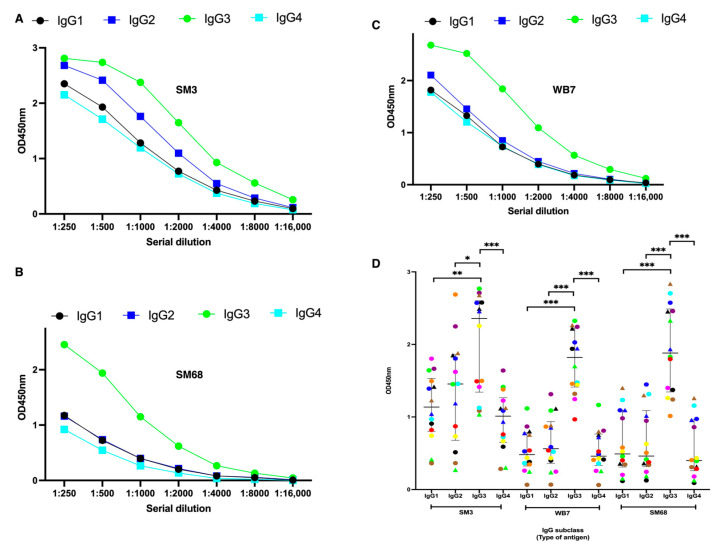
Reactivity assay of the four IgG subclasses to the three candidate biomarkers. (**A**–**C**) The three recombinant antigens were serially diluted and tested for their reactivity by ELISA with the four IgG subclasses to determine the optimal dilution using a pool of 10 highly reacted sera. (**D**) Evaluation of the individual serum (indicated by the specific color code) reactivity of the four IgG subclasses to the determined candidate biomarkers by ELISA. The ELISA results were expressed as mean values ± SD. The *p*-values for the comparison of reactivity between compared groups are indicated. * represents a *p*-value ≤ 0.05, ** represent a *p*-value ≤ 0.01, and *** represent a *p*-value ≤ 0.001.

**Figure 7 ijms-26-05277-f007:**
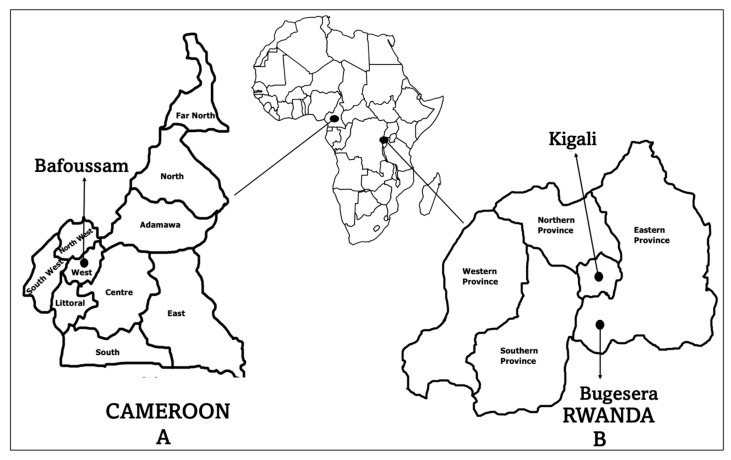
Map of Africa highlighting (**A**) Cameroon and (**B**) Rwanda, along with the locations where blood samples were collected: Bugesera and Bafoussam (healthy individuals) and Kigali (*P. falciparum*-infected symptomatic patients). The magnification of the country maps is not to scale.

**Table 1 ijms-26-05277-t001:** Socio-demographic characteristics of the participants.

Characteristics	Kigali, RWD (n = 75)	Bugesera, RWD (n = 145)	Bafoussam, CMR (n = 120)
Clinical status	Symptomatic	Healthy	Healthy
Gender, n (%)			
Female	35 (46.7)	61 (42.1)	30 (25.0)
Male	40 (53.3)	84 (57.9)	90 (75.0)
Age in years, n (%)			
1–17 (children)	18 (24.0)	2 (1.4)	0 (0.0)
18–49 (adults)	48 (64.0)	127 (87.6)	86 (71.7)
≥50 (elderly)	9 (12.0)	16 (11.0)	34 (28.3)

n = number of participants.

**Table 2 ijms-26-05277-t002:** Primers used in nested PCR for MSP2 allele family specification.

Oligonucleotide	Sequence (5′-3′)
MSP2-Forward	ATGAAGGTAATTAAAACATTGTCTATTATA
MSP2-Reverse	ATATGGCAAAAGATAAAACAAGTGTTGCTG
3D7-Forward	AGTTGAAACATATGAGTATAAGGAGAAGTATG
3D7-Reverse	AATATCTCGAGGGTACTGGTAGATGCTTCTGCATCAT
FC27-Forward	AGTTGAAACATATGAGTATAAGGAGAAGTATGGC
FC27-Reverse	AATATCTCGAGTGCATTGCCAGAACTTGAACTTTCTG

## Data Availability

The raw data supporting the conclusions of this article will be made available by the authors on request.

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
