# Peer review of "Leveraging the Polymorphism of the Merozoite Surface Protein 2 (MSP2) to Engineer Molecular Tools for Predicting Malaria Episodes in a Community"

_ijms, 2025, doi:10.3390/ijms26115277_

Round 1
Reviewer 1 Report
Comments and Suggestions for Authors
The manuscript addresses an important and timely topic—improving malaria surveillance through molecular tools based on Plasmodium falciparum MSP2 polymorphism. The concept of leveraging MSP2 diversity for community-specific malaria risk prediction is novel and highly relevant. However, some methodological aspects, data presentation, and interpretation of results require refinement to enhance the manuscript’s scientific rigor.
The cross-sectional nature of the study limits causal inferences about the predictive capacity of MSP2 polymorphism for malaria episodes. While the authors performed follow-up surveys to assess subsequent malaria episodes, the methodology lacks clarity on how potential confounding factors (e.g., human movement, vector control measures) were accounted for. Including a longitudinal component or statistical adjustments for such variables would strengthen the conclusions.
The selection of MSP2 sequences for antigen design (SM3, WB7, SM68) appears based on antigenicity predictions, but further justification on their representativeness across the entire cohort is needed.
The cloning and expression protocols are adequately detailed, but the rationale behind the failure of other constructs and its potential impact on the findings should be discussed.
The statistical analyses, particularly for the ELISA and survey data, need clearer explanations. The use of p-values without adjusting for multiple comparisons may inflate false-positive rates.
The MOI (Multiplicity of Infection) concept is mentioned but not explicitly linked to the main outcomes. Elaborating on how MOI impacts immune responses or surveillance metrics would be beneficial.
The results from follow-up surveys are intriguing but would benefit from more rigorous analysis. Specifically, quantifying the association between immunity levels and malaria episodes (using ORs or risk ratios) would strengthen the interpretation.
The authors suggest MSP2 polymorphism can predict malaria episodes, yet no clear correlation was established between antibody responses and subsequent malaria incidence. This critical limitation should be acknowledged more transparently.
Integrating computational modeling, as suggested in the abstract, is only briefly mentioned in the discussion without concrete data. Either remove this point or provide supporting details.
Comments on the Quality of English LanguageThe manuscript is generally understandable but requires editing for grammar, sentence structure, and clarity. Some sections are verbose and could be made more concise. Consider a professional English editing service.
Author Response
See attached document

Reviewer 2 Report
Comments and Suggestions for Authors
Kalimba et al. report on investigations regarding the utility of Plasmodium falciparum MSP2 polymorphism as a biomarker for malaria surveillance and outbreak prediction.
Figures 1, 2A, 3, 4A, 4C, 5A, 5B, and 6 are of poor quality. The labels and writing are difficult to read making it difficult to evaluate the data.
In figure 2C there is no band for SDS-PAGE in lane WB7. However, antibody reactivity is shown for WB7 in figures 2D and 4B.
The selection of the three polypeptides reported in section 2.3, SM3, WB7 and SM68 needs further clarification. The region of MSP2 from which the peptides are derived is not clear? Allelic frequency is unclear and similarities or differences to allelic frequencies of MSP2 observed in other African countries is not discussed.
Based on the antibody responses to the three polypeptides in the infected and healthy populations, it is unclear how the peptides can accomplish the goals of serving as biomarkers (the data shown cannot be read) (lines 190-194). Even with the identification of IgG3 as a predominant antibody subclass in assays, the utility of these peptides as biomarkers is in question.
Line 69: "...both gene family are dominant targets..."
Line 115: italicize P. falciparum
Line 119: italicize P. falciparum
Line 153: "(D) The nucleotide and protein sequences of the three antigens" 2D shows western blot data
Author Response
See attached document

Round 2
Reviewer 1 Report
Comments and Suggestions for Authors
I have no further comments
Author Response
Thank you for your valuable review comments
Reviewer 2 Report
Comments and Suggestions for Authors
The authors have answered my queries.
Figures 1A, 3A and 6 are still difficult to read and should be changed.
Author Response
Dear Esteemed Reviewer 2,
The three figures have been revised following a consistent restructuring to enhance clarity and improve readability.
With kindest regards